**Data Availability Statement:** All relevant data are within the manuscript and Supporting Information files.

# The prevalence of ulnar neuropathy at the elbow and ulnar nerve dislocation in recreational wheelchair marathon athletes

**Mari Kakita**[1,2☯]*, **Yukio Mikami**[2☯], **Tatsuru Ibusuki**[2‡], **Takashi Shimoe**[3‡], **Yoshi-ichiro Kamijo**[2☯], **Sven P. Hoekstra**[2,4‡], **Fumihiro Tajima**[2,4☯]

**1** Department of Rehabilitation Medicine, Kansai Electric Power Hospital, Osaka-City, Osaka-Prefecture, Japan, **2** Department of Rehabilitation Medicine, Wakayama Medical University, Wakayama-City, Wakayama-Prefecture, Japan, **3** Department of Orthopedic Surgery, Wakayama Medical University, Wakayama-City, Wakayama-Prefecture, Japan, **4** The Peter Harrison Centre for Disability Sport, Loughborough University, Loughborough, United Kingdom

☯ These authors contributed equally to this work.
‡ These authors also contributed equally to this work.
* sun10139656@yahoo.co.jp

## Abstract

### Background

Ulnar neuropathy at the elbow is an entrapment neuropathy, while ulnar nerve dislocation might also be involved in its incidence and severity. Wheelchair marathon athletes may be at an increased risk for Ulnar Neuropathy. However, there is a paucity of research into the prevalence of Ulnar Neuropathy and ulnar nerve dislocation in this population.

### Objective

To investigate the prevalence of ulnar neuropathy at the elbow and ulnar nerve dislocation in wheelchair marathon athletes.

### Participants

Wheelchair marathon athletes (N = 38) who participated in the 2017, 2018, and 2019 Oita International Wheelchair Marathon. 2 athletes participated only one time, 36 athletes repeatedly. Data from athletes'latest assessment were used.

### Methods

The day before the race, questionnaires, physical examinations, and ultrasonography were conducted to screen for Ulnar Neuropathy in both upper limbs. Ulnar nerve dislocation was confirmed by physical examination and ultrasonography.

### Results

11 (29%) athletes were diagnosed with Ulnar Neuropathy. There were no significant differences in age, height, weight, Body Mass Index, or history of primary illness between athletes with and without Ulnar Neuropathy. In the group without Ulnar Neuropathy, 44% of athletes

**Funding:** The author(s) received no specific funding for this work.

**Competing interests:** The authors have declared that no competing interests exist.

**Abbreviations:** ADL, Activity of Daily Living; BMI, Body Mass Index.

reported to have been engaging in other wheelchair sports, compared to 9% in the group with Ulnar Neuropathy (p = 0.037). Ulnar nerve dislocation was diagnosed in 15 (39%) athletes by ultrasonography. Out of the 14 elbows of 11 athletes diagnosed with Ulnar Neuropathy, 9 (64%) elbows had ulnar nerve dislocation.

## Conclusion

The prevalence of Ulnar Neuropathy in wheelchair marathon athletes was higher than previously reported in able-bodied, non-athlete individuals and lower than in non-athletes with lower limb dysfunction. Therefore, while wheelchair sports may provide some protection against Ulnar Neuropathy, this study further supports the importance of screening for Ulnar Neuropathy, as well as for ulnar nerve dislocation as a potential risk factor for the development of Ulnar Neuropathy.

## Introduction

In recent years, physical activity and sports are increasingly recognized as an invaluable strategy to prevent a sedentary lifestyle in wheelchair users. Moreover, engaging in wheelchair sports can also enhance physical capacity and mental health [1,2]. However, engaging in wheelchair sports places a large mechanical and physiological strain on the upper limbs, potentially leading to upper-body injuries and discontinuation of exercise training [3]. As such, understanding the prevalence of elbow disorders in wheelchair athletes and taking appropriate measures to prevent them is key; not only to maintain the ability to perform exercise training but also for activities of daily living (ADL).

In the able-bodied population, ulnar neuropathy at the elbow is the second most common upper-limb entrapment peripheral neuropathy after carpal tunnel syndrome [4]. Ulnar Neuropathy presents itself at one of two major sites, with 80–85% reported as retro-epicondylar groove [5] and 15–20% as humeroulnar aponeurotic arcade (i.e. the tubular tunnel) [5,6]. Cubital tunnel syndrome is also included in Ulnar Neuropathy, of which the prevalence in the general population is estimated between 1.8–5.9% [7].

Stewart et al [8]. divide the causes of Ulnar Neuropathy into four major factors: 1) external compression, 2) traction during elbow flexion, 3) humeroulnar aponeurotic arcade tension (a tight humeroulnar aponeurotic arcade), 4) a combination of all factors above. There have been many reports about risk factors for the development of Ulnar Neuropathy; these include obesity, being male, smoking and having a job that involves heavy labor using the upper limbs (e.g. farming and manufacturing) [9–13]. Furthermore, it is expected that wheelchair users, who rely on the upper limbs for ADL, sports and physical activity, are also at an increased risk for developing Ulnar Neuropathy compared to able-bodied individuals. To date, however, no studies on the prevalence of Ulnar Neuropathy in non-athlete wheelchair users exist. Nevertheless, the prevalence of Ulnar Neuropathy in lower-limb amputees and people with polio, individuals that are also heavily reliant on their upper body for ADL, is with 70% and 41% respectively higher than in the general population [14,15]. The authors of these studies suggested that unilateral actions such as heavy, repetitive contraction of the flexor capri ulnaris muscle and abnormal elbow mechanics with the use of mobility devices may be contributing factors for Ulnar Neuropathy [14,15]. Although wheelchair propulsion is not a unilateral activity, it does involve a repetitive motion and can place a high strain on the elbow. This indicates that wheelchair users may indeed be at an elevated risk for Ulnar Neuropathy.

Ulnar Neuropathy is also known as a sports-related disorder, and there are case reports on the prevalence of Ulnar Neuropathy in baseball players, Ironman triathletes (associated with the use of aerobars on the bike) and wrestlers [16–18]. Factors related to the development of Ulnar Neuropathy in athletes may differ from those in the general population and include hypertrophy of triceps, the Osborn ligament, and ulnar carpi flexor tendon aponeurosis [19]. In addition, ulnar nerve dislocation during repeated elbow flexion is also suggested as a major cause of Ulnar Neuropathy [20,21]. Although limited in number, previous studies have reported that the prevalence of Ulnar Neuropathy in recreational wheelchair athletes may lie between 25% and 39% [22,23]. However, the prevalence of Ulnar Neuropathy in wheelchair marathon athletes has not yet been studied. The repeated elbow flexion and extension during endurance events such as the wheelchair marathon may place the athletes at a further increased risk for Ulnar Neuropathy.

The present study investigated the prevalence of Ulnar Neuropathy and ulnar nerve dislocation in wheelchair marathon athletes. To inform potential future screening strategies, the relationship between Ulnar Neuropathy and ulnar nerve dislocation as well as personal characteristics was investigated. It was hypothesized that the prevalence of Ulnar Neuropathy was higher in wheelchair marathon athletes compared with able-bodied individuals, and that ulnar nerve dislocation was strongly associated with Ulnar Neuropathy.

## Materials and methods

### Study design

This is a cross-sectional study to investigate the prevalence of Ulnar Neuropathy and ulnar nerve dislocation in wheelchair marathon athletes. On the day before the Oita Wheelchair Marathon, participating wheelchair athletes filled-out questionnaires, while physical and ultrasound examinations were conducted in a dedicated room in a general hospital near the race start to assess symptoms of Ulnar Neuropathy (Fig 1).

The diagnosis of Ulnar Neuropathy was made with a physical examination, while an additional ultrasound examination was used to confirm ulnar nerve partial/complete dislocation.

### Participants

Participants were recruited from the athlete-pool that registered for the 37th (2017), 38th (2018), 39th (2019) Oita International Wheelchair Marathon. The participants of the Oita International Wheelchair Marathon ranged from recreational to professional wheelchair athletes. Exclusion criteria that were applied for participation in the study were a) not speaking Japanese, b) having a cervical spinal cord injury, c) having been previously diagnosed with an upper limb neuromuscular disease other than Ulnar Neuropathy, d) history of trauma or

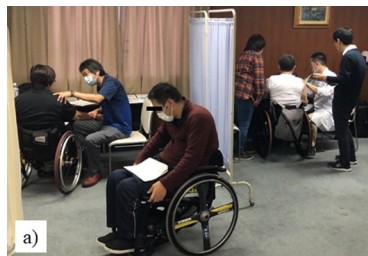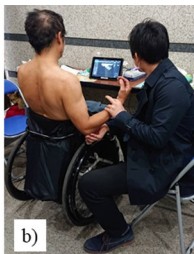

**Fig 1. Ulnar Neuropathy screening for wheelchair marathon athletes.** (a), Physical examination; (b), Ultrasound examination.

surgery at the elbow joint within 6 months prior to the race. The study was approved by the Research Ethics Review Committee of the Wakayama Medical University (#2735) and each participant provided written informed consent before enrolment into the study and for publication. The individual captured in Figs 1 and 2 has given written informed consent (as outlined in the PLOS consent form).

## Questionnaires

The bespoke questionnaires used included questions on age, gender, height, weight, body mass index (BMI), history of primary illness, the presence of comorbidities, history of sports injury, dominant hand, number of wheelchair marathons completed, history of engagement in other sports, types and volume of exercise training performed in a representative week, symptoms such as pain and numbness in one of the upper limbs, the timing and location of these symptoms, history of consultation and treatment of the symptoms in the upper limb; and were filled-out by the participants themselves.

## Physical examination

In total, four rehabilitation physicians performed the physical examination, where one physician examined both upper limbs of one athlete. With the participant in a sitting position, Froment's sign, paralysis or muscular atrophy of the first dorsal interosseous muscle or abductor of the little finger, Tinel's sign at the cubital tunnel (the specificity is 48–100% and the sensitivity is 44–75% [24]), numbness of the ulnar nerve region, and ring split sign were examined. Subjective numbness was assessed using a questionnaire; with a distinction made between transient and persistent numbness in the ulnar nerve region. Athletes who had any the following outcomes were diagnosed as having Ulnar Neuropathy: a) Constant or transient numbness of the ulnar nerve region or positive for ring split sign, b) Positive for Tinel's sign at the cubital tunnel, c) Positive for Froment's sign, or paralysis / muscle atrophy of the first dorsal interosseous muscle or little finger abductor muscle [25]. Ulnar nerve dislocation was also examined by moving the participant's elbow from full extension to maximum flexion. Ulnar nerve dislocation was accepted when the ulnar nerve climbed up the medial epicondyle and moved from the ulnar nerve groove during this motion (Fig 2A and 2B) [25].

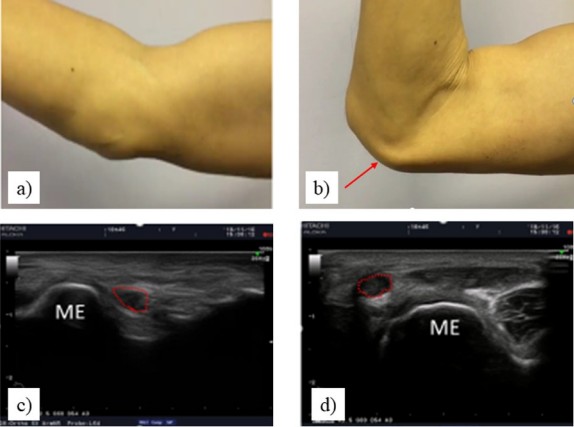

**Fig 2. The detection of ulnar nerve dislocation.** (a, b), Physical examination findings (c, d), Ultrasound examination findings. ME, medial epicondyle.

During the physical examination, the participant's elbow was moved from (a) full extension to (b) maximum flexion. As shown by the arrow, ulnar nerve dislocation was accepted when the ulnar nerve climbed up the medial epicondyle and moved from the ulnar nerve groove. Similarly, the participant's elbow was moved from (c) the maximum extension position to (d) the maximum flexion position in the ultrasound examination. Diagnosis was deemed "non-injured" when the ulnar nerve did not cross the medial epicondyle. Partial dislocation was diagnosed when the ulnar nerve remained at the apex of the medial epicondyle. If the ulnar nerve surrounded by red circle completely passed over the medial epicondyle, it was diagnosed as complete dislocation.

## Ultrasound examination

Ulnar nerve dislocation was diagnosed by ultrasound examination. Dynamic ultrasound has been demonstrated to have a diagnostic accuracy of 89 to 100% [26]. Two experienced orthopedic surgeons participated in the ultrasound examination, and one surgeon examined both upper limbs for one participant. A LOGIQ e Premium, (GE Healthcare, Chicago, Illinois) Doppler ultrasound machine with a 12 MHz probe was used for the complete examination. In B mode, the ulnar nerve was identified by the medial epicondyle in a uniaxial view. The participant's elbow was then moved from the maximum extension position to the maximum flexion position. Partial dislocation was diagnosed when the ulnar nerve remained at the apex of the medial epicondyle. If the ulnar nerve completely passes over the medial epicondyle, it was diagnosed as complete dislocation (Fig 2C and 2D) [25].

## Statistical analysis

Continuous variables were expressed as mean ± SD. Outcomes of the questionnaire and the presence of ulnar nerve dislocation were compared between the group with Ulnar Neuropathy and the group without Ulnar Neuropathy (non-Ulnar Neuropathy), using unpaired t-tests for continuous variables and chi-square tests for nominal variables. The statistical software JMP® pro version 14.1 (SAS Institute Inc.) was used for all analyses and p<0.05 was accepted as statistically significant.

## Results

A total of 77 athletes were screened (n = 33 in 2017, n = 20 in 2018 and n = 24 in 2019). Of these, athletes a with cervical spinal cord injury (n = 9 in 2017, n = 7 in 2018, n = 8 in 2019) were excluded, and the results of the most recent screening were used for athletes who were screened more than once. Finally, 38 participants (n = 20 in 2017; n = 10 in 2018; n = 8 in 2019) were included in the analysis. A total of 76 upper limbs, two per athlete, were examined. Most participants were male (n = 36), while spinal cord injury was the most common disability (n = 29). Table 1 shows the participants'characteristics. The age was 50 ± 15 yrs, the time since injury 29 ± 15 yrs and BMI 22 ± 2.9 kg/m2.

11 athletes (28.9%), 14 limbs (18%), were diagnosed as having Ulnar Neuropathy. Ulnar Neuropathy was found on the non-dominant arm of 3 athletes, on the dominant arm of 5 athletes, in both arms in 3 athletes. The physical examination revealed numbness in 4 athletes; 1 athlete reported constant numbness, 3 athletes transient numbness. Two athletes out of the 3 reported transient numbness after exercise. Three athletes reported ring split sign, while 10 reported Tinel's sign at the cubital tunnel. No motor impairment such as paralysis or muscle atrophy in the upper body was observed in any of the athletes (Table 2).

Tables 1 and 3 show the personal characteristics and training as well injury history of the athletes with and without Ulnar Neuropathy. There were no significant differences in

**Table 1. Comparisons of demographic characteristics between non-Ulnar Neuropathy athletes and Ulnar Neuropathy athletes.**

|  | non-Ulnar Neuropathy(N = 27) | Ulnar Neuropathy(N = 11) | p value |
|---|---|---|---|
| Age (yrs) | 49.85±16.88 | 53.09±9.45 | 0.56 |
| Male (%)/Female (%) | 96.2/3.7 | 100/0 | - |
| Body height (cm) | 166.40±10.55 | 168.25±3.37 | 0.60 |
| Body weight (kg) | 62.75±9.99 | 63.1±8.34 | 0.92 |
| Body Mass Index | 22.70±2.91 | 22.28±2.87 | 0.71 |
| History of primary disease (yrs) | 26.0±12.25 | 32.18±17.84 | 0.40 |

Data are mean ± SD.

*, independent t-test, p<0.05, non-Ulnar Neuropathy group VS Ulnar Neuropathy group.

age, height, weight, BMI, or history of primary illness between athletes with and without Ulnar Neuropathy (p>0.40). In both groups, most athletes were right-handed. The average history of participation in wheelchair marathons was 17 years, while 13 athletes (34%) was also engaging other sports. In the group without Ulnar Neuropathy, 44% of athletes reported to have been engaging in other wheelchair sports, compared to 9% in the group with Ulnar Neuropathy (p = 0.037). In total, 12 athletes (31%) reported to regularly engage in strength training, without a difference between both groups (p = 0.98) (Table 3).

Complete ulnar nerve dislocation was observed in 8 athletes (21%) (4 on the dominant side, 2 on the non-dominant side, 2 on both sides), and partial dislocation was observed in 9 athletes (23.6%) (2 on the dominant side, 6 on the non-dominant side, 1 on both sides). Ulnar nerve dislocation was observed in 9 out of 14 elbows with Ulnar Neuropathy (64%), significantly more than in elbows without Ulnar Neuropathy (p = 0.0003) (Fig 3).

The prevalence of ulnar nerve partial/complete dislocation was significantly higher in the group with Ulnar Neuropathy than in the group without Ulnar Neuropathy.

**Table 2. Physical signs and symptoms in the 11 athletes with Ulnar Neuropathy.**

| ID | A) numbness | B) ring split sign | C) | D) | Ulnar Neuropathy |
|---|---|---|---|---|---|
| 1 | + | - | + | - | left (non-dominant) |
| 2 | + | - | - | - | right (dominant) |
| 3 | - | + | + | - | left (dominant) |
| 4 | - | - | + | - | left (non-dominant) |
| 5 | - | - | + | - | right (non-dominant) |
| 6 | - | - | + | - | right (dominant) |
| 7 | - | + | + | - | right (dominant) |
| 8 | + | - | + | - | left (non-dominant) |
| 9 | - | + | + | - | bilateral |
| 10 | - | - | + | - | bilateral |
| 11 | + | - | + | - | bilateral |

A) numbness.

B) ring split sign.

C) Tinel' sign at the cubital tunnel.

D) Froment's sign or weakness or atrophy of 1st dorsal interosseous muscle.

+sign, positive.

-sign, negative.

**Table 3. Comparisons of training history and personal characteristics between non-Ulnar Neuropathy athletes and Ulnar Neuropathy athletes.**

|  | non-Ulnar Neuropathy (N = 27) | Ulnar Neuropathy (N = 11) | p value |
|---|---|---|---|
| Right dominant hand N (%) | 23 (85.1) | 10 (90.9) | - |
| Duration of wheelchair marathon (years) | 16.51±11.07 | 19.68±9.65 | 0.42 |
| History of other wheelchair sports (%) | 12 (44.4) | 1 (9.0) | 0.037* |
| History of wheelchair sports injury (%) | 9 (33.3) | 3 (27.2) | 0.10 |
| Training frequency (times/week) | 3.14±1.69 | 2.57±1.58 | 0.37 |
| Training time (hours/session) | 1.44±0.56 | 1.61±0.45 | 0.40 |
| Resistance training (%) | 8 (29.6) | 4 (36.3) | 0.98 |

Continuous variables were expressed as the mean ± SD.

*, independent t-test, p<0.05, non-Ulnar Neuropathy group VS Ulnar Neuropathy group

## Discussion

This is the first study to investigate the prevalence of Ulnar Neuropathy at the elbow and ulnar nerve dislocation in wheelchair marathon athletes. The prevalence of Ulnar Neuropathy in wheelchair marathon athletes was 28.9%, which was higher than in able-bodied persons but lower than reported in other populations with lower limb dysfunction [7,14,15]. In addition, the prevalence was comparable to that reported in wheelchair athletes engaging in track racing and intermittent team sports [22,23].

These findings are in line with our hypothesis. The reason for the higher prevalence of Ulnar Neuropathy in wheelchair athletes compared with able-bodied individuals is expected to be largely related to the use of a wheelchair as a mode of propulsion. However, as there are limited studies on Ulnar Neuropathy in wheelchair users, the magnitude of the increased risk and the mechanisms explaining this phenomenon remain unknown. Nonetheless, several factors associated with wheelchair racing may be implicated. During wheelchair propulsion, the pressure within the cubital tunnel increases 7-fold with elbow flexion and more than 20-fold when contraction of the flexor carpi ulnaris muscle is added [27]. Goosey reported that during

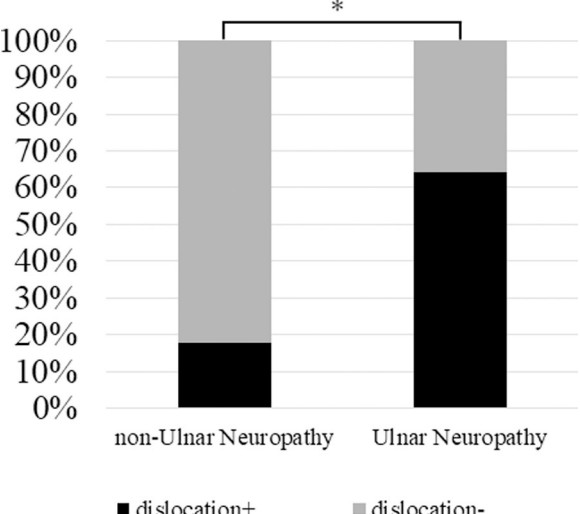

**Fig 3. The prevalence of ulnar nerve partial/complete dislocation in Ulnar Neuropathy group and non-Ulnar Neuropathy group.** *, χ square test, p<0.05, non-Ulnar Neuropathy group VS Ulnar Neuropathy group.

wheelchair racing, the elbow maximumly flexes from 168 to 180˚at 14mph [28]. While these factors may theoretically place wheelchair athletes at a further heightened risk for Ulnar Neuropathy compared with non-athletic wheelchair users, the prevalence of Ulnar Neuropathy in wheelchair marathon athletes was *lower* than found in non-athletes with lower limb dysfunction [7,14,15].

The lower prevalence of Ulnar Neuropathy in the wheelchair athletes in the present and other studies [22,23] compared with non-athlete populations with lower limb deficiencies [14,15] suggests that engaging in regular exercise may provide some protection against the development of Ulnar Neuropathy. This effect may be related to several factors. For instance, obesity, Diabetes Mellitus and smoking have been reported as risk factors for Ulnar Neuropathy [10,29]. Although there is large inter-individual variability in weight loss following regular exercise training [30], it is conceivable that the ~4.5 h training/week of the wheelchair athletes included in the present study has contributed to the maintenance of what is considered a healthy body mass. Furthermore, athletes without Ulnar Neuropathy engaged more in other sports apart from wheelchair marathon racing compared with the group with Ulnar Neuropathy. It may be that simultaneously engaging in other sports provides a more holistic form of conditioning, which, considering the role of skeletal muscle imbalances as a risk factor for sports injuries [31], may provide additional protection against Ulnar Neuropathy. In contrast, there was no difference in the time the athletes spend on resistance training between the groups with and without Ulnar Neuropathy. Finally, it should be noted that this cross-sectional study cannot exclude the possibility that the lower prevalence of Ulnar Neuropathy in wheelchair athletes compared with non-athlete populations with a disability is not the result of the exercise training but rather the result of a higher physical capacity or less severe disability *per se*, enabling the participants in the present study to engage in sports while this may not be an option for those with other disabilities [14,15].

It has been reported that the pathology of Ulnar Neuropathy varies depending on the underlying cause, and the treatment strategies should be adapted accordingly [32]. For instance, when the complaints occur at the retro-epicondylar groove, often found in the non-dominant hand and caused by external compression, conservative treatment such as the release of compression is recommended. On the other hand, humeroulnar aponeurotic arcade is often found in the dominant hand of people repeatedly engaging in heavy physical labor and the main cause is thought to be nerve entrapment. This type of Ulnar Neuropathy is considered more severe, and surgery is recommended as soon as possible after diagnosis. Most cases of Ulnar Neuropathy are associated with the retro-epicondylar groove. While this is a less severe from Ulnar Neuropathy, early detection may be crucial to prevent aggravation of the symptoms. In the present study, only three athletes had subjective symptoms such as numbness, and eight others were diagnosed with Ulnar Neuropathy for the first time by this screening. None of athletes with Ulnar Neuropathy showed any motor impairments and all were able to complete the wheelchair marathon, suggesting that the detection of Ulnar Neuropathy does not necessarily interfere with sport engagement and can be managed when diagnosed at an early stage.

In this study, partial and complete dislocation were observed in 23.6% and 21% of all participants by ultrasound examination, respectively. This prevalence is comparable to that reported in the general population [25,33]. In studies investigating members of the general population, partial nerve dislocation was present in 6.9% to 25.3% of the Ulnar Neuropathy cases, while complete nerve dislocation was present in 11.3% to 21.1% of the Ulnar Neuropathy cases [25,33]. In the present study, partial and complete nerve dislocation was observed in 14% and 50% of the Ulnar Neuropathy cases, respectively. As such, although its exact role

in the aetiology of Ulnar Neuropathy remains to be established, ulnar nerve dislocation is strongly associated with Ulnar Neuropathy. Kang et al. [25] showed that persons with Ulnar Neuropathy and ulnar nerve dislocation had larger axonal damage, suggesting that this may mediate the effect of nerve dislocation on the development of Ulnar Neuropathy. Therefore, it is recommended to include examinations of ulnar nerve dislocation in future Ulnar Neuropathy screening programs.

Despite reported associations between Ulnar Neuropathy and ulnar nerve dislocation, it remains debated whether the latter indeed causes Ulnar Neuropathy. For instance, Omejec et al. [34] found that ulnar nerve dislocation tended to be more common in controls compared with Ulnar Neuropathy patients. The reason for the equivocal findings on the role of ulnar nerve dislocation may be the existence of two major different pathological conditions (related to the humeroulnar aponeurotic arcade and retro-epicondylar groove) in Ulnar Neuropathy. As such, it is necessary to examine the predictive value of partial and complete ulnar nerve dislocation separately. Omejec et al. [34] reported that in patients with Ulnar Neuropathy at the humeroulnar aponeurotic arcade, the frequency of partial dislocation tended to be lower and complete dislocation higher compared to the cases of Ulnar Neuropathy at the retro-epicondylar groove. It was hypothesized that entrapment under the humeroulnar aponeurotic arcade prevents the ulnar nerve from gliding along the elbow during maximal flexion. As a result, ulnar nerve traction forces the nerve to shortcut the medial epicondyle and completely dislocate [33]. Leis et al. [35] investigated 133 patients with complete ulnar nerve dislocation to determine whether this condition is a risk factor for Ulnar Neuropathy. They found that Ulnar Neuropathy occurs less frequently and is less severe on the side of complete dislocation and concluded that complete dislocation is not associated with a higher incidence of Ulnar Neuropathy and that it may actually have a protective effect on the ulnar nerve [35]. The results of the present study showed that the prevalence of complete dislocation of Ulnar Neuropathy was particularly high at 50%, but further studies are needed to determine its role in Ulnar Neuropathy in wheelchair racers.

Although the prevalence of Ulnar Neuropathy seems lower in wheelchair athletes compared with non-athletes with a disability, the higher prevalence compared with the general population suggests that regular, large-scale Ulnar Neuropathy screening in wheelchair athletes may be considered in the future. Indeed, as engagement in exercise training is highly important for the maintenance of physical fitness, health and the ability to successfully perform ADL, such screening methods may aid to prevent long-term interruption of exercise training. As part of these screening programs, this study has underscore the usefulness of ultrasonography, making it possible to diagnose Ulnar Neuropathy cases with ulnar nerve dislocation which were difficult to evaluate by electrophysiological examinations only. Future studies could build on the findings and techniques presented here by investigating strategies to prevent Ulnar Neuropathy in wheelchair athletes. Although, for preventing Ulnar Neuropathy, it is also necessary to clarify the risk, but in this study, it was not clear. For find out whether wheelchair marathon or sports may cause Ulnar Neuropathy, Ulnar Neuropathy in non-athletic manual wheelchair users need to be investigated in the future study. Heyward proposed that modelling studies should be performed in order to biomechanically and systematically investigate, evaluate and dissect load during athletic tasks in their systematic review about shoulder complaints in Wheelchair athletes, biomechanical modelling studies [36]. We agree with them. Exploring the relationship between training load and the development of the condition in Ulnar Neuropathy, or studying the significance of wheelchair propulsion technique is required in the future study.

## Study limitations

As with any study, the present study had some limitation and provides opportunities for future research. First, as this study was conducted in wheelchair athletes taking part in a wheelchair marathon the following day, it may be that the athletes included in the present study were only those in a good condition and not those in which injuries interfered with participation in the wheelchair marathon. Second, while this is the first study to investigate the prevalence of Ulnar Neuropathy in wheelchair marathon athletes, future studies may benefit from a larger sample size. This may enhance the statistical power to investigate the influence of personal characteristics and training regimes on Ulnar Neuropathy risk. In addition, as most of the participants in this study were male, translation of the present findings to females should be done with caution. Future studies should aim to replicate this study in a cohort of female wheelchair athletes. Off-season screening and/or including screening in additional events may be strategies to include more athletes in such studies. Third, as the screening was performed by four different physicians, there may have been variability between them with regards to the detection of Ulnar Neuropathy and ulnar nerve dislocation. Finally, as the screening was performed as part of a large, international wheelchair marathon event, and study set-up had to be adjusted accordingly, electrophysiological examinations were not feasible to include in the present study. Considering the high specificity of electrophysiological examinations in Ulnar Neuropathy detection, this could be considered as a follow-up from on-site screening as performed in the current study.

## Conclusion

The lower prevalence of Ulnar Neuropathy in wheelchair marathon athletes compared with non-athletes with lower-limb deficiencies suggests that regular exercise may be protective against the development of Ulnar Neuropathy. Nonetheless, the prevalence is higher compared with that reported in the general population. As such, regular screening for Ulnar Neuropathy may be recommended to protect wheelchair athletes from training interruptions caused by upper-limb injuries. Considering its strong association with Ulnar Neuropathy, examinations for the detection of ulnar nerve dislocation should be part of such a screening program.

## Supporting information

**S1 Fig. This is the questionnaires in English.**
(PDF)

**S2 Fig. This is the questionnaires in Japanese which we used in the study.**
(PDF)

**S1 Table. This is the minimum data set No.1.**
(PDF)

**S2 Table. This is the minimum data set No.2.**
(PDF)

## Author Contributions

**Conceptualization:** Mari Kakita.

**Data curation:** Mari Kakita.

**Formal analysis:** Yukio Mikami.

**Investigation:** Mari Kakita, Yukio Mikami, Tatsuru Ibusuki, Takashi Shimoe, Yoshi-ichiro Kamijo, Sven P. Hoekstra.

**Project administration:** Yoshi-ichiro Kamijo.

**Resources:** Mari Kakita, Takashi Shimoe.

**Supervision:** Yukio Mikami, Fumihiro Tajima.

**Visualization:** Mari Kakita.

**Writing – original draft:** Mari Kakita.

**Writing – review & editing:** Yukio Mikami, Sven P. Hoekstra.

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
