## [Decision Letter · Decision Letter 0]

18 Sep 2020

PONE-D-20-19978

The prevalence of ulnar neuropathy at the elbow and ulnar nerve dislocation in recreational wheelchair marathon athletes

PLOS ONE

Dear Dr. Kakita,

Thank you for submitting your manuscript to PLOS ONE. After careful consideration, we feel that it has merit but does not fully meet PLOS ONE’s publication criteria as it currently stands. Therefore, we invite you to submit a revised version of the manuscript that addresses the points raised during the review process.

Please make sure to address both reviewers' comments, especially regarding the abstract and abbreviations.

We look forward to receiving your revised manuscript.

Kind regards,

Benjamin A. Philip

Academic Editor

PLOS ONE

Journal Requirements:

2. Thank you for stating that “The study was approved by the Research Ethics Review Committee of the University (#2735)”. Please ensure that the full name of the ethics committee which approved this study is stated, both in the methods section of your manuscript and on the online submission.

4. We note that Figure 1 includes an image of a participant in the study. 

Reviewers' comments:

Reviewer's Responses to Questions

**Comments to the Author**

1. Is the manuscript technically sound, and do the data support the conclusions?

Reviewer #1: Yes

Reviewer #2: Yes

2. Has the statistical analysis been performed appropriately and rigorously? 

Reviewer #1: Yes

Reviewer #2: Yes

3. Have the authors made all data underlying the findings in their manuscript fully available?

Reviewer #1: Yes

Reviewer #2: Yes

4. Is the manuscript presented in an intelligible fashion and written in standard English?

Reviewer #1: Yes

Reviewer #2: Yes

5. Review Comments to the Author

Reviewer #1: Overall, this is a well-thought and interesting study presenting novel information.

Lines 65-70. Clarify if all prevalence numbers are for the able-bodied population.

Lines 143-146. Clarify if this was performed using ultrasound. It is unclear if ulnar dislocation was also assessed by physical exam alone. Include sensitivity and specificity information where appropriate.

Methods section: Clarify physical exam criteria for UNE. Was numbness subjective or objective based on exam? Was it constant vs transient numbness, a history of numbness or current symptoms? And is there any information in prior literature regarding the sensitivity or specificity of Tinel’s sign, ring split sign or Froment’s sign?

Lines 224-228. Rather than provide information regarding “right” or “left” findings, would say “dominant” vs “non-dominant” to make it more meaningful to the reader.

2 areas to provide more references:

1. To strengthen the discussion section, include information about wheelchair propulsion techniques as a potential cause of UNE as evaluation of mechanics would often be the first treatment for UNE specifically in wheelchair athletes. Authors should look into literature on wheelchair configuration and push mechanics especially as it pertains to marathon wheelchair athletes. No such articles cited.

2. Greater literature review needed on the correlation of UNE with ulnar dislocation in able-bodied population. There are multiple, conflicting studies on this. Prior literature suggests that ulnar dislocation actually protects against UNE findings, or no correlation between UNE and ulnar dislocation, and others suggest more severe ulnar neuropathy with ulnar dislocation. Your findings are interesting, that ulnar dislocation was associated with positive findings of UNE.

Reviewer #2: The authors address an important topic, Ulnar Neuropathy at the elbow, relevant for the health of wheelchair users. I have a few minor comments that I hope improve the quality of the paper

Abstract:

- I would prefer an abstract without abbreviations

- From the abstract it is not directly clear if the 38 participant are unique or repeated over the severel years of measurement

- The results are also not clear from the abstract

Intro:

line 57 Non-commmunicable, perhaps some, surely not all, perhaps more about preventing a sedentary lifestyle, consider revising

Line 64 insert for as in 'but also for'

The paper is very descriptive in nature. There is a hypothesis, which would be good to get back to in the discussion.

Methods:

117 what is meant with randomly

124 from which University

177, consider using effect sizes as well?

184, perhaps this can be put in abstract

185 most participants.., I would say almost all are male, is this also relevant as a limitation?

195 I find this paragraph a bit confusing and the table clearer, perhaps put table above text and refer to it earlier. Perhaps also sort the table as dominant, non-dominant, bilateral

UNE- and UNe + I would say UNE and non-UNE

244 the comparison with refs 14 and 15 seems rather important, perhaps you can get a bit more detail of their results and numbers and how they compare to yours.

247 DM, did not see that defined before.

Perhaps the discussion would benefit with a 'so what ' or 'what next' answer. What do your findings mean for wheelchair users, are there any advises to be given? Or could you tell us how to go on from here and what kind of study would tell us more about the relation of UNE and doing sports, etc.

General comment of a personal nature, I dislike all the abbrevations actually, especially reading the discussion, they are not intuitive to me and make need to go back and forth a lot. I think it is not really a problem to spell out the terms that are rather self-explanatory. Also if you just call it an Ulnar Neuropathy throughout, I think most will understand you talk about the elbow.

6. PLOS authors have the option to publish the peer review history of their article (what does this mean?). If published, this will include your full peer review and any attached files.

Reviewer #1: No

Reviewer #2: **Yes: **dr. Riemer JK Vegter

---

## [Author Response · Author response to Decision Letter 0]

29 Oct 2020

PONE-D-20-19978

The prevalence of ulnar neuropathy at the elbow and ulnar nerve dislocation in recreational wheelchair marathon athletes

Responses to the Editor and Reviewers

Journal Requirements: When submitting your revision, we need you to address these additional requirements. 

Response: 

We created the manuscript according to the PLOS ONE style templates and ensured that the manuscript meets the PLOS ONE’s style requirements.

2. Thank you for stating that “The study was approved by the Research Ethics Review Committee of the University (#2735)”. Please ensure that the full name of the ethics committee which approved this study is stated, both in the methods section of your manuscript and on the online submission. 

Response: 

The full name of the ethics committee is the Research Ethics Review Committee of the Wakayama Medical University; approved No. 2735, which has been included both in the Methods section of the manuscript (Lines 133-135) and on the additional information for online submission.

Response: 

The questionnaire of this study in Japanese and English has been added as Supporting Information (supplement).

4. We note that Figure 1 includes an image of a participant in the study. As per the PLOS ONE policy (http://journals.plos.org/plosone/s/submission-guidelines#loc-human-subjects-research) on papers that include identifying, or potentially identifying, information, the individual(s) or parent(s)/guardian(s) must be informed of the terms of the PLOS open-access (CC-BY) license and provide specific permission for publication of these details under the terms of this license. Please download the Consent Form for Publication in a PLOS Journal (http://journals.plos.org/plosone/s/file?id=8ce6/plos-consent-form-english.pdf). The signed consent form should not be submitted with the manuscript , but should be securely filed in the individual's case notes. Please amend the methods section and ethics statement of the manuscript to explicitly state that the patient/participant has provided consent for publication: “The individual in this manuscript has given written informed consent (as outlined in PLOS consent form) to publish these case details”. If you are unable to obtain consent from the subject of the photograph, you will need to remove the figure and any other textual identifying information or case descriptions for this individual. 

Response: 

We downloaded the Consent Form for Publication in a PLOS Journal and obtained consent for publication from the subjects shown in Figure 1. We amended the Methods section and ethics statement of the manuscript: “each participant provided written informed consent before enrolment into the study and for publication. The individual captured in Fig. 1 and 2 has given written informed consent (as outlined in the PLOS consent form).” was added. (Lines 135-137)

5. In your Data Availability statement, you have not specified where the minimal data set underlying the results described in your manuscript can be found. PLOS defines a study's minimal data set as the underlying data used to reach the conclusions drawn in the manuscript and any additional data required to replicate the reported study findings in their entirety. All PLOS journals require that the minimal data set be made fully available. For more information about our data policy, please seehttp://journals.plos.org/plosone/s/data-availability. Upon re-submitting your revised manuscript, please upload your study’s minimal underlying data set as either Supporting Information files or to a stable, public repository and include the relevant URLs, DOIs, or accession numbers within your revised cover letter. For a list of acceptable repositories, please see http://journals.plos.org/plosone/s/data-availability#loc-recommended-repositories. Any potentially identifying patient information must be fully anonymized. Important: If there are ethical or legal restrictions to sharing your data publicly, please explain these restrictions in detail. Please see our guidelines for more information on what we consider unacceptable restrictions to publicly sharing data: http://journals.plos.org/plosone/s/data-availability#loc-unacceptable-data-access-restrictions. Note that it is not acceptable for the authors to be the sole named individuals responsible for ensuring data access. We will update your Data Availability statement to reflect the information you provide in your cover letter. 

Response: 

We uploaded the minimal data set underlying the results described in our manuscript as Supporting Information (supplement).

Response: 

ORCID ID of corresponding author, Mari Kakita is (0000-0003-4384-6156)

Reviewers' comments:

Reviewer's Responses to Questions Comments to the Author

1. Is the manuscript technically sound, and do the data support the conclusions?

Reviewer #1: Yes, Reviewer #2: Yes

2. Has the statistical analysis been performed appropriately and rigorously? 

Reviewer #1: Yes, Reviewer #2: Yes

3. Have the authors made all data underlying the findings in their manuscript fully available?

Reviewer #1: Yes, Reviewer #2: Yes

4. Is the manuscript presented in an intelligible fashion and written in standard English?

Reviewer #1: Yes, Reviewer #2: Yes

5. Review Comments to the Author

Dear Reviewer 1, 

Thank you for your kind words about our manuscript and the insightful comments. We especially appreciate the suggestions you have provided to strengthen the discussion around methodological aspects and the potential causes of ulnar neuropathy at the elbow. Please find below our responses to each individual comment.

Reviewer #1: Overall, this is a well-thought and interesting study presenting novel information.

Lines 65-70. Clarify if all prevalence numbers are for the able-bodied population.

Response: All prevalence numbers are indeed for the able-bodied population. “In the able-bodied population” was added in the manuscript. (Line 68)

Lines 143-146. Clarify if this was performed using ultrasound. It is unclear if ulnar dislocation was also assessed by physical exam alone. Include sensitivity and specificity information where appropriate.

Response: Ulnar nerve dislocation was observed by both physical and ultrasound examinations, but the diagnosis was made by ultrasound examination for more accuracy.

In the Physical examination section of Methods, “Ulnar nerve dislocation was diagnosed when”, has been corrected to “Ulnar nerve dislocation was accepted when” (Lines 168-169).

Also, in the ultrasound examination section of the Methods, “Dynamic ultrasound has been demonstrated to have a diagnostic accuracy of 89 to 100%. [26]” (Lines177-178). “Ulnar nerve dislocation was diagnosed by ultrasound examination.” was added (Line 177).

In the Discussion ” by ultrasound examination” was added to “In this study, partial and complete dislocation were observed in 23.6% and 21% of all participants, respectively” (Line 314).

Methods section: Clarify physical exam criteria for UNE. Was numbness subjective or objective based on exam? Was its constant vs transient numbness, a history of numbness or current symptoms? And is there any information in prior literature regarding the sensitivity or specificity of Tinel’s sign, ring split sign or Froment’s sign?

Response: “Subjective numbness was assessed using a questionnaire, and it was confirmed that there was transient or persistent numbness in the ulnar nerve region.” was added in the Physical examination section of Methods (Lines 152-54). And “Constant or transient numbness” was also added (Line 155). 

With regards to Tinel’s sign, “the specificity is 48-100% and the sensitivity is 44-75%. [24]” was added in the manuscript (Lines 150-151). We are not aware of any studies that have investigated the sensitivity or specificity of ring split signs and Fromant’s signs.

Lines 224-228. Rather than provide information regarding “right” or “left” findings, would say “dominant” vs “non-dominant” to make it more meaningful to the reader.

Response: We have corrected all the descriptions of “right” and “left” to “dominant” and “non-dominant”.

2areas to provide more references:

1.To strengthen the discussion section, include information about wheelchair propulsion techniques as a potential cause of UNE as evaluation of mechanics would often be the first treatment for UNE specifically in wheelchair athletes. Authors should look into literature on wheelchair configuration and push mechanics especially as it pertains to marathon wheelchair athletes. No such articles cited.

Response: Based on your advice, we have added the following considerations to the discussion (Lines 268-72):

“Nonetheless, several factors associated with wheelchair racing may be implicated. During wheelchair propulsion, the pressure within the cubital tunnel increases 7-fold with elbow flexion and more than 20-fold when contraction of the flexor carpi ulnaris muscle is added.[27] Goosey reported that during wheelchair racing, the elbow maximumly flexes from 168 to 180°at 14mph.[28]”

2. Greater literature review needed on the correlation of UNE with ulnar dislocation in able-bodied population. There are multiple, conflicting studies on this. Prior literature suggests that ulnar dislocation actually protects against UNE findings, or no correlation between UNE and ulnar dislocation, and others suggest more severe ulnar neuropathy with ulnar dislocation. Your findings are interesting, that ulnar dislocation was associated with positive findings of UNE.

Response: We agree that the role of ulnar nerve dislocation is controversial. As such, we have included a separate paragraph in the Discussion to give this aspect more attention from Lines 326-347. 

Dear Reviewer 2, 

Thank you for your kind words about our manuscript and the expert comments you have provided. We particularly value your comments with regards to the practical implications of the presented work. We believe they have improved the quality of the manuscript. Please find below the responses to each individual comment. 

Reviewer #2: The authors address an important topic, Ulnar Neuropathy at the elbow, relevant for the health of wheelchair users. I have a few minor comments that I hope improve the quality of the paper

Abstract:

- I would prefer an abstract without abbreviations

Response: The abstract is modified and does no longer include abbreviations. 

- From the abstract it is not directly clear if the 38 participants are unique or repeated over the several years of measurement

Response: We have added the following sentences in the abstract (Lines 36-38), “Wheelchair marathon athletes (N=38) who participated in the 2017, 2018, and 2019 Oita International Wheelchair Marathon. 2 athletes participated only one time, 36 athletes repeatedly. Data from athletes` latest assessment were used.”

- The results are also not clear from the abstract

Response: We modified the abstract to let the Results section stand out clearer.

Intro:

line 57 Non-communicable, perhaps some, surely not all, perhaps more about preventing a sedentary lifestyle, consider revising

Response: We corrected the sentence “to prevent non-communicable disease” to “to prevent a sedentary lifestyle” in the Introduction section (Line 61).

Line 64 insert for as in 'but also for'

The paper is very descriptive in nature. There is a hypothesis, which would be good to get back to in the discussion.

Response: We revised the sentences to “not only to maintain for the ability to perform exercise training but also for activities of daily living (ADL)” (Line 66-67). 

We agree that we could have addressed the hypothesis in more detail in the Discussion. We have now included a new paragraph on the link between Ulnar Neuropathy and ulnar nerve dislocation (Lines 326-347), while we have also discussed the role of wheelchair propulsion in Ulnar Neuropathy in more detail at the start of the Discussion to address the other aspect of the hypothesis (Lines 263-275). 

Methods:

117 what is meant with randomly

Response: The meaning of randomly was that we did not request specific athletes to take part. However, since we acknowledge that this statement can cause confusion, we have removed “randomly” in the manuscript (Line 127).

124 from which University

Response: We have changed this to “Wakayama Medical University” (Lines 134-135).

177, consider using effect sizes as well?

Response: As this was a cross-sectional study, we did not consider using effect sizes and preferred to limit ourselves to the statistics presently used.

184, perhaps this can be put in abstract

Response: We have now included this in the Abstract (Line 36-38).

185 most participants..., I would say almost all are male, is this also relevant as a limitation?

Response: We have added the following sentence in the Study Limitations (Lines 378-380): “In addition, as most of the participants in this study were male, translation of the present findings to females should be done with caution. Future studies should aim to replicate this study in a cohort of female wheelchair athletes.”

195 I find this paragraph a bit confusing and the table clearer, perhaps put table above text and refer to it earlier. Perhaps also sort the table as dominant, non-dominant, bilateral

UNE- and UNe + I would say UNE and non-UNE

Response: We have now placed the table above the text. We have corrected all the descriptions of “right” and “left” to “dominant” and “non-dominant” and “UNE- “and “UNE+” to “UNE” and “non-UNE”.

244 the comparison with refs 14 and 15 seems rather important, perhaps you can get a bit more detail of their results and numbers and how they compare to yours.

Response: We have added the following sentences in the Introduction. (Lines 92-98): “The authors of these studies suggested that unilateral actions such as heavy, repetitive contraction of the flexor capri ulnaris muscle and abnormal elbow mechanics with the use of mobility devices may be contributing factors for UNE [14,15]. Although wheelchair propulsion is not a unilateral activity, it does involve a repetitive motion and can place a high strain on the elbow. This indicates that wheelchair users may indeed be at an elevated risk for Ulnar Neuropathy.” 

247 DM, did not see that defined before.

Response: Based on your valid general comment regarding the use of abbreviations, we have now removed the abbreviation for Diabetes Mellitus. (Line 280)

Perhaps the discussion would benefit with a 'so what ' or 'what next' answer. What do your findings mean for wheelchair users, are there any advises to be given? Or could you tell us how to go on from here and what kind of study would tell us more about the relation of UNE and doing sports, etc.

Response: Based on your advice, we have tried to better address the practical implications of the present findings and include recommendations for future studies in the Discussion. This is now discussed in Lines 357-368.

General comment of a personal nature, I dislike all the abbreviations actually, especially reading the discussion, they are not intuitive to me and make need to go back and forth a lot. I think it is not really a problem to spell out the terms that are rather self-explanatory. Also, if you just call it an Ulnar Neuropathy throughout, I think most will understand you talk about the elbow.

Response: We have now removed the use of UNE and have used Ulnar Neuropathy throughout the manuscript.

6. PLOS authors have the option to publish the peer review history of their article (what does this mean?). If published, this will include your full peer review and any attached files.

Do you want your identity to be public for this peer review? For information about this choice, including consent withdrawal, please see our Privacy Policy. Reviewer #1: No, Reviewer #2: Yes: dr. Riemer JK Vegter

---

## [Decision Letter · Decision Letter 1]

19 Nov 2020

The prevalence of ulnar neuropathy at the elbow and ulnar nerve dislocation in recreational wheelchair marathon athletes

PONE-D-20-19978R1

Dear Dr. Kakita,

We’re pleased to inform you that your manuscript has been judged scientifically suitable for publication and will be formally accepted for publication once it meets all outstanding technical requirements.

Kind regards,

Benjamin A. Philip

Academic Editor

PLOS ONE

Additional Editor Comments (optional):

Reviewers' comments:

Reviewer's Responses to Questions

**Comments to the Author**

1. If the authors have adequately addressed your comments raised in a previous round of review and you feel that this manuscript is now acceptable for publication, you may indicate that here to bypass the “Comments to the Author” section, enter your conflict of interest statement in the “Confidential to Editor” section, and submit your "Accept" recommendation.

Reviewer #1: All comments have been addressed

Reviewer #2: All comments have been addressed

2. Is the manuscript technically sound, and do the data support the conclusions?

Reviewer #1: Yes

Reviewer #2: Yes

3. Has the statistical analysis been performed appropriately and rigorously? 

Reviewer #1: Yes

Reviewer #2: Yes

4. Have the authors made all data underlying the findings in their manuscript fully available?

Reviewer #1: Yes

Reviewer #2: Yes

5. Is the manuscript presented in an intelligible fashion and written in standard English?

Reviewer #1: Yes

Reviewer #2: No

6. Review Comments to the Author

Reviewer #1: (No Response)

Reviewer #2: The authors did a good job in responding to my comments, I am happy to accept the paper in its current form.

7. PLOS authors have the option to publish the peer review history of their article (what does this mean?). If published, this will include your full peer review and any attached files.

Reviewer #1: **Yes: **Dr Andrea Cyr

Reviewer #2: **Yes: **Riemer JK Vegter

---

## [Editor Report · Acceptance letter]

23 Nov 2020

PONE-D-20-19978R1 

The prevalence of ulnar neuropathy at the elbow and ulnar nerve dislocation in recreational wheelchair marathon athletes 

Dear Dr. Kakita:

I'm pleased to inform you that your manuscript has been deemed suitable for publication in PLOS ONE. Congratulations! Your manuscript is now with our production department. 

Kind regards, 

on behalf of

Dr. Benjamin A. Philip 

Academic Editor

PLOS ONE